# Does Eating-Away-from-Home Increase the Risk of a Metabolic Syndrome Diagnosis?

**DOI:** 10.3390/ijerph16040575

**Published:** 2019-02-16

**Authors:** Hui Wang, Yingjie Yu, Xu Tian

**Affiliations:** 1Department of Epidemiology and Biostatistics, School of Public Health, Nanjing Medical University, Nanjing 210011, China; huiwang@njmu.edu.cn; 2Department of Nutrition and Food Hygiene, Beijing Research Center for Preventive Medicine, Beijing 100034, China; yyj.jane.1982@163.com; 3College of Economics and Management, China Center for Food Security Studies, Nanjing Agricultural University, Nanjing 210095, China

**Keywords:** eating-away-from-home, metabolic syndrome, metabolic syndrome components, sex

## Abstract

Rising frequency of eating-away-from-home (EAFH) is suspected to be correlated with several non-communicable diseases. This study adopted the Chinese Health and Nutrition Survey (CHNS) 2009 data to investigate the association between being diagnosed with the metabolic syndrome (MetS) and EAFH at different ages. Results showed that the association between EAFH and MetS varied at different ages and differed for males and females. EAFH was positively associated with a higher risk of getting MetS for males, especially for those aged between 45 and 60; while it was negatively associated with the risk of getting MetS for young females (<45) (all *p* < 0.05). In particular, EAFH was associated with a lower risk of getting high serum triglycerides (TGs), abdominal adiposity, elevated blood pressure, and impaired fasting blood glucose for young females, while higher risk of high serum TGs, abdominal adiposity, elevated blood pressure, and impaired fasting blood glucose for middle-aged males (all *p* < 0.05). In addition, a higher frequency of EAFH was associated with a higher risk of abdominal adiposity and elevated blood pressure for older women, and a lower risk of elevated blood pressure, and impaired fasting blood glucose for younger men (all *p* < 0.05). Our study implies that heterogeneous target strategies for preventing MetS in different subpopulation should be considered.

## 1. Introduction

The successful economic development and rapid urbanization in the past decades have greatly reshaped the dietary habits of Chinese people. In particular, consumers preferred to eat more meals in restaurants, cafeterias, and street vendors. The National Bureau of Statistics of China (NBSC) showed that the total turnover of catering industry increased from 52.9 billion Chinese yuan (RMB) in 2000 to 512.7 billion in 2016. Previous studies have also claimed that the share of food expenditure on eating-away-from-home (EAFH) raised from 7.9% in 1992 to 21.2% in 2010, in urban China [1,2], and from 2.3% to 13.3% in rural China [3], which could be attributed to the soaring frequency of eating in restaurants and other food suppliers [4]. 

This shifting dietary habit is associated with an epidemiological transition [5], which is categorized by a high prevalence of coronary heart disease, certain types of cancer, obesity, and non-insulin-dependent diabetes [6,7]. In particular, the soaring prevalence of metabolic syndrome (MetS) had caused rising concern worldwide. MetS is a complex of anthropometric and biochemical marker aberrations, which includes abdominal obesity, insulin resistance, dyslipidemia, and high blood pressure [8]. MetS represents an important risk factor for developing cardiovascular disease (CVD) and type 2 diabetes (T2D) [9], and affects more than 25% of the world population [10]. CVD is the leading cause of death, worldwide, except Africa, which resulted in 17.9 million deaths and accounted for 32.1% of global deaths in 2015. Meanwhile, approximately 382 million people had the T2D worldwide, of which 175 million cases were undiagnosed in 2013. The prevalence of CVD and T2D went up sharply, over the past few decades, and was expected to keep rising in the next few decades [11,12]. China is one of the countries which had a high prevalence of CVD and T2D. Approximately 290 million individuals in China suffer from CVD, which accounted for more than 40% of deaths in both rural and urban areas, in 2013 [13]. The prevalence of diabetes in adults also reached 12% in 2013, of which 50% had prediabetes [14].

Inappropriate diet was believed to be a leading risk factor of non-communicable diseases like multiple chronic diseases [15] and obesity [4,16]. Due to differences in preparation methods, ingredients, and recipes, food prepared away from home is usually categorized as high-calorie density, deep-oiled, low micronutrient intake, and higher in sodium and fat, compared with dishes prepared at home [4,17,18,19,20]. As a consequence, EAFH is suspected to be one of the culprits of the rising prevalence of several non-communicable diseases, like cardiovascular disease and obesity [1,21]. However, EAFH also increases the availability and diversity of diet for consumers [4,22], which in turn plays a protective role against several diseases like obesity and metabolic syndrome (MetS), as well [16,23].

Previous literature has highlighted the importance of a healthy balanced diet and physical activity, for preventing MetS. A more diversified diet, if it characterized by increasing consumption of fruits, vegetables, and dairy products, will play a protective function in the pathogenesis of CVD and T2D [24,25] and further contribute to reducing adiposity and cardio-metabolic health [26,27]. However, if more high-calorie-density foods are consumed, due to increased enjoyability of new food items [22,26], a greater dietary variety could also lead to the rising prevalence of obesity [16]. In addition, eating alone [28] and EAFH were also detected to be associated with a higher risk of MetS [1,4]. 

In general, most studies focused on the association between EAFH and obesity, or dietary diversity and health, while little is known about the impact of EAFH on MetS. We, have thus, investigated the association between frequency of EAFH and MetS, using Chinese Health and Nutrition Survey (CHNS) data.

## 2. Materials and Methods

### 2.1. Population and Data Collection

CHNS was a longitudinal household survey, jointly conducted by the University of North Carolina at Chapel Hill (UNC-CH) and the Chinese Institute of Nutrition and Food Safety (INFS), and China Center for Disease Control and Prevention (CCDC). This survey was approved by the Institutional Review Board of these institutions. All participants provided written informed consent and all methods were performed in accordance with the relevant guidelines and regulations. Detailed information about this survey and laboratory examinations has been described elsewhere [29]. In addition, the ethics committees of the Medical Faculty of the University of Goettingen and the University of North Carolina at Chapel Hill approved our use of these data, in 2013 (Application no.: 26/6/13 An). This study only adopted the 2009 data as the biochemical data was only collected in this year.

Only adults (≥18 years old) with complete information on food consumption were included in our analysis (n = 5372). We excluded adults with implausible daily energy intake (>7000 kcal or <520 kcal) (n = 4). Women who were pregnant or lactating (n = 39), adults who have a history of metabolic related disease, such as myocardial infarction (n =19), diabetes (n = 90), and apoplexy (n = 30), and people who take antihypertensive drugs (n = 542) were excluded because their diet might have changed after the diagnosis of the disease. In addition, 121 individuals who provided incomplete information and 9 individuals whose household income was negative were deleted, as well. Finally, 4518 participants were included in the present study (2441 females and 2077 males). The selection of sample is demonstrated in a flow chart in the Appendix A.

### 2.2. Dietary Assessment and EAFH

Individual daily food consumption data were collected through face-to-face interviews for three consecutive days, which were randomly distributed within one week. Trained field interviewers helped each participants recall the food that they consumed at home and outside, in a 24h period, and recorded the codes (listed in the Food Composition Table of China), types, amounts, and locations of consumption for each food item, by using food models and pictures. More detailed information about the dietary data in the CHNS has been previously reported [29].

For each meal (breakfast, lunch, dinner, and snack) in each day, the place of consumption was reported. In this study, EAFH were defined as all meals that were not consumed at home, during the three survey days, including meals purchased at restaurants, fast food outlets, cafeterias, and other venues, such as food stands. It also included meals that were free, hosted by friends or relatives, or were provided by work units. We counted the number of meals ate-away-from-home, during the three days, which ranged from 0 to 12 (including snacks). Furthermore, the frequency of EAFH was categorized into three groups: never (0), sometimes (>0 and <=3), and often (>3 and <=12).

### 2.3. Biomarker Variables and MetS

Waist circumference was measured around the body at the top of the hipbone, using an unstretched tape over the light cloth. Measurement was conducted without putting any pressure to body surface, and the value was recorded to the nearest 0.1 cm. Weight and height of each individual were measured by trained health workers, using regularly calibrated equipment (SECA880 scales and SECA 206 wall-mounted metal tapes). BMI was calculated as the ratio of weight (kg) divided by the square of height (m2). Blood pressure was measured after three rest in a seated position, each rest lasted for 5 minutes. It was utilized to calculate the average value of systolic blood pressure (SBP) and diastolic blood pressure (DBP). Blood sample was collected after 12–14 h overnight fasting from all participants, and was stored in vacationer tubes. All blood samples were analyzed in the central laboratories of the China–Japan Friendship Hospital. Fasting plasma glucose was measured by the glucose oxidase-peroxidase (GOD-PAP), using the kit produced by Randox, UK. Serum high-density lipoprotein cholesterol (HDL-C) concentration was measured by enzymatic method; serum triglycerides (TGs) levels was measured by the cholesterol-peroxidase (CHOD-PAP); these two testing reagents were produced by the Kyowa, Japan individually. The last three indices were measured by the Hitachi 7600 machine.

MetS was defined according to the harmonized definition of the International Diabetes Federation and other organizations [8], three or more out of five following criteria were considered as MetS: (1) central adiposity (WC ≥90 cm in men and ≥85 cm in women), which was adjusted according to the criteria of central obesity of Chinese adults [30]; (2) serum HDL-C <50 mg/dL in women or <40 mg/dL in men; (3) serum TGs levels >150 mg/dL; (4) SBP ≥130 mmHg or DBP ≥85 mmHg; and (5) fasting plasma glucose ≥100 mg/dL.

### 2.4. Measurement of Other Covariates

Information about sex, age, income per capita, and educational level (primary, middle, and high) can be found in a previous study [16], total energy intake, smoking status (1 = currently smoking; 0 = otherwise) and drinking status (1 = drank alcohol in the past year; 0 = otherwise), and residential area (urban/rural and north/south) were collected as co-variables. Physical activity was defined according to occupation (1 = light physical activity, working in a sitting or standing position like office worker and teacher; 2 = moderate physical activity, such as student or driver; 3 = heavy physical activity, such as farmers, loader, miner), and adjusted in the model as well. Adults were categorized into three age groups, according to the WHO standards (young: ≥18 and ≤45; middle aged: >45 and ≤60; old: >60).

### 2.5. Statistical Analysis

Descriptive statistics are presented at different frequencies of EAFH, for each sex. Continuous variables are presented as means ± SEs and categorical variables are presented as percentages.

The association between EAFH and MetS and its individual components was detected by a multivariable-adjusted logistic regression, both the OR values and 95% CI are presented (CI in brackets). Sex, income per capita, educational level, physical activity, age, smoking and drinking status, fat share, total energy intake, and residential location were adjusted in the multivariable regression model. Income and total energy intake were measured in logarithm. The association between predicted probability of MetS and number of meals EAFH was mapped using kernel-weighted local polynomial smoothing; both the fitted probability and 95% confidence interval are presented. The linear trend of odds ratio for EAFH was tested by using number of meals EAFH in the multivariable logistic regression.

All analyses were performed with Stata MP 14.0 (Stata Crop, USA) and *p* < 0.05 was considered as statistically significant.

## 3. Results

### 3.1. Summary of Population Characteristics

Mean and S.D. of individual characteristics at each group of EAFH are presented in Table 1. Female adults who had a higher frequency of EAFH were younger and richer, similar trends were also detected for males (all *p* < 0.001). Meanwhile, both males and females with middle and high education level were more likely to eat away from home (all *p* < 0.001). Conversely, people who were engaged in heavy activity were less likely to eat away from home (all *p* < 0.05). Female smokers were less likely to eat away from home (*p* = 0.003), while male smokers and drinkers were more likely to eat outside of family (*p* = 0.003). In addition, rural residence and northern people had a lower frequency of EAFH, compared to their counterparts (all *p* < 0.001). Strikingly, females who ate outside of family had a slightly lower BMI (*p* = 0.004), but no such evidence was detected for males (*p* = 0.486).

### 3.2. Nutrients Intake and MetS

Table 2 reports the nutrients intake and metabolic risk factors of participants by EAFH frequency and sexes. Interestingly, females who ate away from home, frequently, had a significantly lower intake of total energy (*p* = 0.008), which could be attributed to lower intake of carbohydrate (*p* < 0.001); but their protein intake was higher than those who never ate outside of home (*p* =0.024). Males who ate frequently away from home also had a significant lower intake of carbohydrate (*p* < 0.001), but higher intake of fat and protein (*p* = 0.020 and *p* < 0.001). These trends were consistent with the energy share of each macro nutrients at different frequencies of EAFH. Lower waist circumference was observed for females who ate away from home more frequently (*p* = 0.006), while an insignificant positive association was found for males (*p* = 0.137). The TGs, fasting blood glucose, SBP and DBP decreased significantly with a higher frequency of EAFH for women (all *p* < 0.05), while only a significant negative association between SBP and frequency of EAFH was detected for men (*p* = 0.002). We also found that women who ate away from home frequently, had a lower risk of getting MetS (*p* = 0.002), which could be attributed to a lower serum TG levels, lower abnormal glucose homeostasis, and a lower blood pressure (all *p < 0.05*). In contrast, men who ate away from home frequently, had a higher risk of high serum TGs levels and low HDL-C (*p* = 0.005).

### 3.3. Association between EAFH and MetS and its Components

Further investigation was conducted to investigate the association between frequency of EAFH and MetS odds (Table 3). We found that people who ate away from home frequently had a significantly higher risk of getting MetS [odds ratio and 95% CI, 1. 475(1.121, 1.942) and 1.678(1.149, 2.451)], while the positive association was only detected in males, once we ran the regression by gender, separately. In particular, the odds of males who ate at least once and more than three times during the three-day period was 38.3% and 50.0% higher than those who never ate outside [1.383(1.043, 1.834), 1.500(1.023, 2.199)]. 

We also mapped the association of the predicted risk of getting MetS and the frequency of EAFH in Figure 1. We found an uptrend for males but a U-trend for females. Moreover, the risk of getting MetS varied for different age and gender groups.

A significant gender difference was detected between females and males. In particular, females who ate away from home at least once and more than three times during the three-day period had significantly lower risk of getting MetS than their male counterparts [0.622(0.428, 0.904), 0.443(0.247, 0.795)]. The risk of getting MetS also varied greatly over different age groups. We found that the risk of getting MetS was significantly higher for middle-aged females [2.524(1.952, 3.263)], old females [2.968(2.153, 4.091)], and middle-aged males [1.476(1.140, 1.910)]. The predicted probability of getting MetS, for each gender and age cohort, is presented in Figure 2. To test the robustness of our results, we adopted the number of meals ate outside of home to measure the frequency of EAFH, and used age and age-squared to replace the two age dummy variables. Results are presented in Appendix A. A significant positive association between EAFH and the risk of getting MetS was confirmed but was only significant for males. Similarly, the risk of getting MetS followed an inverse-U shape trend for both genders, and peaked at around 50 for males and 75 for females, respectively (see Appendix A in Appendix A). More importantly, females had a lower risk of being MetS, compared with their male counterparts before 50 years old, but they were more vulnerable to have MetS after getting older.

Finally, the associations between the five individual components of MetS and EAFH were investigated, separately, in a multivariable adjusted logistic regressions, in both sexes and different categories of ages (see Table 4). A significant negative association between EAFH and the risk of getting MetS was detected for young females, while a positive association was found for middle-aged and older males (all *p* < 0.01). The lower probability of having MetS for young females could be attributed to the declining risk of high serum TGs, abdominal adiposity, elevated blood pressure, and impaired fasting glucose at a higher frequency of EAFH (all *p* < 0.05). In contrast, old females had a higher risk of abdominal adiposity and elevated blood pressure once they ate outside of home more frequently (all *p* < 0.05). Different from females, a higher risk of high serum TGs, abdominal adiposity, elevated blood pressure, and impaired fasting glucose was associated with EAFH frequency for middle-aged males, and older males also had a higher risk of elevated blood pressure if they often ate away from home (all *p* < 0.05). Intriguingly, young males who ate frequently away from home had a higher risk of low HDL, but a lower risk of elevated blood pressure, and impaired fasting glucose (all *p* < 0.05).

In light of the vast heterogeneous cuisine across different regions in China, we also investigated the association between EAFH and MetS, separately, by regions. Appendix A presented the main results for rural/urban and south/north, respectively. In general, similar results were detected in each region, but no significant association between EAFH and MetS was found in Northern China.

## 4. Discussion

This study found a negative association between EAFH and Mets, in young females, which could be attributed to lower serum TGs, lower abdominal adiposity, lower blood pressure, and a lower impaired fasting glucose. However, a positive association was observed in middle-aged males, which could be explained by the higher serum TGs, abdominal adiposity, elevated blood pressure, and impaired fasting glucose. In addition, a positive association was also detected in older males, which was mainly caused by the increasing risk of elevated blood pressure. 

Our results revealed that EAFH contributed to the high serum TGs and impaired fasting glucose in middle-aged men, significantly, while a reverse association was detected in young females. These findings were partially consistent with the results revealed in previous studies [20,31,32]. One study claimed that men were susceptible to a higher BMI and waist circumference with a higher frequency of EAFH, but the association between body mass and EAFH were not always significant for women [31]. It is reasonable that adults’ biomarkers alter before they became overweight or obesity. The conflicting results between men and women can be attributed to three facts. First, males and females are biologically different, and have very different gender psychosocial stressors and lifestyle; for instance, the adipocytes functions impacted by the conversion of testosterone to estradiol in women and men and the concentration of leptin is higher in women than in men [33,34]. Second, women have more knowledge about a balanced diet, since most women dominate the kitchen and are responsible for the diet of the whole family in China [31]. Third, Chinese men engaged in more social activities, which usually take place in restaurants (see Appendix A), and they tend to drink (see Table 1) and eat a lot, during these meals [4]. On the contrary, females, particularly young females, were usually more self-aware and more subject to peer pressure and had a higher preferences for thinness [35]; thus they were less likely to be affected by EAFH. More importantly, young females who had a higher frequency of EAFH were usually those who had a higher social-economic-status, and they were well-educated (see Table 1), had decent jobs (which can be reflected by the lower physical activity level shown in Table 1), and had a healthier lifestyle (less likely to smoke and drink as shown in Table 1), all of which could contribute to a lower risk of MetS. Previous studies have also found that EAFH contributed to a higher BMI for Chinese men, while it was not associated with a BMI increase for women, and they even found some evidence that women who eat lunch away from home more often had a lower BMI [4]. Our results demonstrated that women preferred food which is high in protein but low in carbohydrate, when eating outside, while men preferred to eat food higher in fat and protein. Our previous research found that women would like to choose a combination of low-energy density but nutrient-rich diet, along with increasing dietary diversity. Differently, men would like to choose a diet which is high in energy but nutrient-poor like deep fried or braised with soy sauce meat.

The most interesting finding of the present study is that the negative association was only observed in young females, which was in line with studies working on the associations between snacking behaviors or dietary diversity and BMI, carried out in female university students [36,37]. Both of those two studies indicated that female students were prone to choose healthier snacks and dietary patterns to control their body weight. Additionally, the level of fasting glucose and blood pressure of young males were negatively related with the frequency of eating out. Therefore, it is possible that young people got used to new healthy dietary guidelines more easily, which is evidenced in many countries under nutrition transition. A study carried out in Korea claimed that young people who live alone tend to prefer simple food [28]. 

Several limitations should be mentioned in the current study. First, our findings are drawn from a cross-sectional data, thus, causality cannot not be claimed and remains for future studies with longitudinal data. Second, EAFH is defined as all food prepared outside of home. However, the preparing methods vary greatly across the location of EAFH. For instance, food prepared in a school canteen might be very different from those prepared in full-service restaurants. Those heterogeneities might also affect the association between EAFH and MetS. Third, accompanying people who ate away from home along with respondents might also affect the variety of diet and further influence the risk of getting MetS, which however, has not been controlled in the present paper, due to data availability. Fourth, companies or institutions in China typically offer free or low-cost lunches to employees during working hours, for use at company-owned restaurants, cafeterias, and food shops. Generally, these meals were lower energy density than those prepared in full-service restaurants. However, we could not distinguish these meals from each other, which remains for future research with high-quality data. This situation was similar with the research carried out by Eric Robinson, which claimed that salad meals offered by full-service restaurants had significantly more calorie in comparison with the one offered by fast food brand chains, such as Burger King, KFC, and McDonalds [38]. Although no research has investigated the calorie levels of full-service restaurants in China, yet, restaurants regularly add more salt, oil, or energy dense foods to attract customers. Furthermore, China has experienced various changes in demographic and health-related characteristics in the past nine years, while CHNS has only collected biomarker data for 2009, thus, new dataset was needed to evaluate our finding. 

## 5. Conclusions

In conclusion, the present study emphasizes the disparate associations between EAFH and MetS and its components in Chinese females and adult males. Middle-aged males were prone to get MetS when eating out frequently, while young females were more likely to reduce their risk of getting MetS when eating out very often. Regarding the fast, economic development and increasing popularity of modern urban living style in China, EAFH will become increasingly popular, which might lead to a rising prevalence of MetS. Plenty of evidences indicated that prevention is much more cost-effective in comparison to clinical treatment. Results from our study indicates that heterogeneous target strategies for preventing MetS in different subpopulation should be considered. Moreover, food labeling which provides information on nutrients of food and consequence of MetS, could also play a role in practice.

## Figures and Tables

**Figure 1 ijerph-16-00575-f001:**
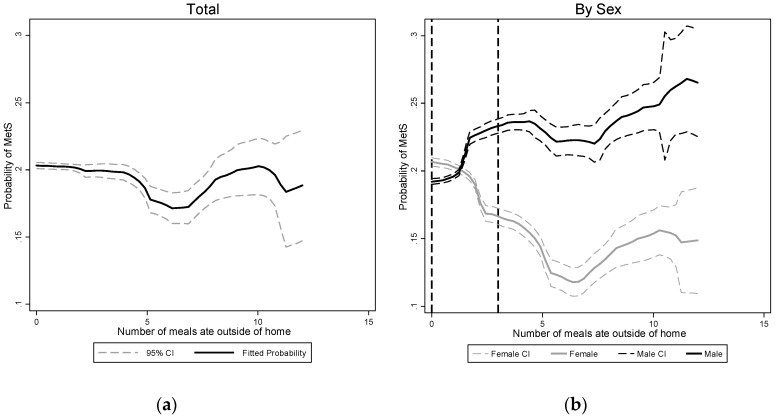
Associations between predicted probability of having metabolic syndrome diagnosis and EAFH. (**a**) Association for the whole sample; (**b**). Associations for female and male respectively. Note: The solid black line indicates the fitted probability of having metabolic syndrome, and the two dash lines indicate the 95% CI of the fitted probability. The two dashed vertical lines refer to the two cut-off points of frequency of EAFH (0 and 3). Fitted probabilities were estimated using logistic regression, after adjusting for age, educational level (primary, middle, and high), logarithm of income, smoking (yes/no), drinking (currently drinking or not), physical activity (light, moderate, and heavy), localization (urban or rural; north or south), total energy intake and fat share. For total population regression, sex and its interaction term with frequency of EAFH was also added as covariates.

**Figure 2 ijerph-16-00575-f002:**
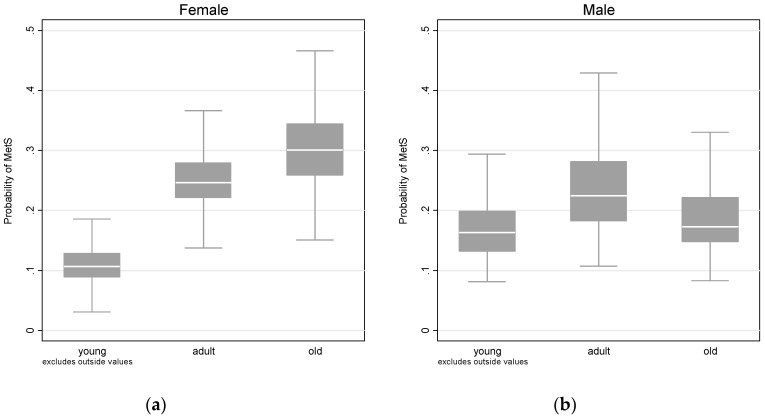
Predicted probability of having a metabolic syndrome diagnosis, at each age cohort. (**a**) Association for female (**b**) Association for male. Note: The box indicates the 75% and 25% percentiles, and the white line within the box indicates the median value. Two grey horizontal lines refer to the upper and lower adjacent values. Young is defined as ≤45 years; adult is defined as >45 y and ≤60 years; old is defined as >60 years.

**Table 1 ijerph-16-00575-t001:** Characteristics of participants by frequency of EAFH ^1^ and sexes.

	Females				Males			
	Never	Sometimes	Often	*p* Trend ^2^	Never	Sometimes	Often	*p* Trends
Observations	1798	476	167		1449	431	197	
Age(y)	50.2 ±14.2	46.7 ± 14.3	40.4 ± 13.3	<0.001	50.9 ± 15.1	47.2 ± 14.4	42.9 ± 13.2	<0.001
ln(income) ^3^	10.0 ±1.4	10.2 ± 1.3	10.3 ± 1.0	<0.001	10.0 ± 1.4	10.2 ± 1.4	10.3 ± 1.4	<0.001
Educational level(%) ^4^								
primary	52.1	34.9	22.8	<0.001	35.7	24.6	18.3	<0.001
middle	44.5	58.2	62.9	<0.001	59.1	67.3	68.5	<0.001
high	3.3	6.9	14.4	<0.001	5.2	8.1	13.2	<0.001
Physical activity(%) ^5^								
light	27.7	35.1	34.1	0.002	23.2	28.1	22.8	0.364
middle	24.9	28.4	37.7	<0.001	16.9	22.5	22.8	0.004
heavy	47.4	36.6	28.1	<0.001	59.9	49.4	54.3	0.002
Smoking(%)	3.9	2.3	0.0	0.003	54.7	60.1	64	0.004
Drinking(%)	7.6	12.4	7.8	0.059	59.1	68.9	70.1	<0.001
Rural (%)	75.3	49.4	47.9	<0.001	76.9	54.8	50.8	<0.001
North(%)	49.2	22.1	29.9	<0.001	48.6	26.2	21.8	<0.001
BMI	23.3 ± 3.5	23.0 ± 3.3	22.5 ± 3.2	0.004	23.1 ± 3.3	22.9 ± 3.1	23.4 ± 3.8	0.486
BMI ≥ 28(%)	10.0	8.0	7.2	0.098	7.1	4.9	12.2	0.184

Notes: 1. Never, sometimes, and often refer to the frequency of EAFH: never (0), sometimes (>0 and <=3), and often (>3 and <=12). Continuous variables are presented as mean ± SD and categorical variables are presented as percentages. 2. *p* trends were analyzed by chi-square trends analysis. 3. ln(income) means logarithm of income per capita per year. 4. Education level (primary school education, middle school education, or above high school education). 5. Physical activity (according to the occupation type, ranging from 1–5; this classification conforms to the original data from the CHNS:1 = very light physical activity, working in a sitting position (e.g., office worker or watch repairer); 2 = light physical activity, working in a standing position (e.g., sales person or teacher); 3 = moderate physical activity (e.g., student or driver); 4 = heavy physical activity (e.g., farmer or dancer); and 5 = very heavy physical activity (e.g., loader, logger, or miner); 1 and 2 are classified as light activity, 3 are classified as moderate activity, 4 and 5 are classified as heavy activity).

**Table 2 ijerph-16-00575-t002:** Dietary intake and metabolic risk factors of participants by EAFH frequency ^1^ and sexes.

	Females				Males			
	Never	Sometimes	Often	*p* Trend ^2^	Never	Sometimes	Often	*p* Trends
Observations	1798	476	167		1449	431	197	
Nutrients								
Total energy(kcal/day)	1992 ± 585	1971 ± 562	1852 ± 588	0.008	2339 ± 663	2402 ± 666	2254 ± 715	0.611
Carbohydrate(g/day)	281 ± 96	260 ± 89	245 ± 83	<0.001	329 ± 106	313 ± 106	291 ± 99	<0.001
Fat(g/day)	68 ± 32	75 ± 33	68 ± 40	0.052	79 ± 38	88 ± 39	80 ± 51	0.020
Protein(g/day)	61 ± 21	63 ± 21	64 ± 22	0.024	70 ± 23	77 ± 26	77 ± 25	<0.001
Energy share (%)								
Carbohydrate	56.6	52.9	54.0	<0.001	56.5	52.5	52.8	<0.001
Fat	30.9	34.1	31.7	<0.001	30.2	32.3	30.8	0.007
Protein	12.3	12.8	14.1	<0.001	12.1	12.9	13.9	<0.001
Mets Markers								
Waist circumference (cm)	80 ± 10	80 ± 10	78 ± 9	0.006	83 ± 10	83 ± 9	85 ± 10	0.137
Serum triglycerides (mg/dL)	133 ± 101	125 ± 104	119 ± 96	0.043	147 ± 149	162 ± 142	162 ± 125	0.054
HDL-C (mg/dL)	57 ± 15	57 ± 14	59 ± 18	0.377	55 ± 17	53 ± 16	53 ± 17	0.054
Fasting blood glucose (mg/dL)	95 ± 21	93 ± 12	90 ± 11	0.001	95 ± 23	95 ± 25	94 ± 20	0.368
Systolic blood pressure (mmHg)	121 ± 17	117 ± 17	115 ± 15	<0.001	125 ± 16	122 ± 16	122 ± 15	0.002
Diastolic blood pressure (mmHg)	78 ± 10	76 ± 10	76 ± 10	<0.001	81 ± 10	80 ± 10	81 ± 11	0.205
Share of patients								
MetS ^3^ (%)	21.9	17.6	13.8	0.002	18.4	22.5	22.3	0.053
Abdominal adiposity (%)	32.4	30.9	25.8	0.092	26.3	26.0	30.5	0.350
High serum triglyceride level (%)	27.6	23.1	19.2	0.003	30.0	36.4	37.1	0.005
Low HDL-C (%)	31.4	31.3	33.5	0.690	13.9	18.1	17.3	0.047
Abnormal glucose homeostasis (%)	23.1	20.0	15.6	0.012	25.9	23.0	24.4	0.326
Elevated blood pressure (%)	32.6	24.4	15.0	<0.001	41.8	35.7	39.6	0.119

Notes: 1. Never, sometimes, and often refer to the frequency of EAFH: never (0), sometimes (>0 and <=3), and often (>3 and <=12). Continuous variables are presented as mean ± SD and categorical variables are presented as percentages. 2. *p* trends were analyzed by chi-square trends analysis. 3. MetS was defined as the presence of three or more of the following components: (1) abdominal adiposity (WC ≥102 cm in men and ≥ 88 cm in women; (2) low serum HDL-cholesterol < 40 mg/dL for men and <50 mg/dL for women); (3) high serum triglyceride levels (>150 mg/dL); (4) elevated blood pressure (SBP ≥130 mmHg or DBP ≥ 85 mmHg); and (5) abnormal glucose homeostasis (fasting plasma glucose level ≥100 mg/dL).

**Table 3 ijerph-16-00575-t003:** Multivariable adjusted ^1^ association between Mets and EAFH.

	Total Population (n = 4518)	Females (n = 2441)	Males (n = 2077)
Never(referent)	1	1	1
Sometimes	1.475(1.121, 1.942) ^2^	0.962(0.727, 1.274)	1.383(1.043, 1.834)
Often	1.678(1.149, 2.451)	0.861(0.535, 1.385)	1.500(1.023, 2.199)
Females	1.172(0.933, 1.472)		
Females*sometimes	0.622(0.428, 0.904)		
Females*often	0.443(0.247, 0.795)		
Middle aged	1.942(1.616, 2.332)	2.524(1.952, 3.263)	1.476(1.140, 1.910)
Elderly adults	1.861(1.469, 2.358)	2.968(2.153, 4.091)	1.085(0.768, 1.534)

Notes: 1. Adjusted for age category (young: ≥18 & ≤45; middle aged: >45 & ≤60; elderly adults: >60), educational level (primary, middle, and high), ln (income), smoking (currently smoking or not), drinking (currently drinking or not), physical activity (light, moderate, and heavy), localization (urban or rural; north or south), total energy intake and fat share. For total population regression, sex was added in addition. 2. Values are ORs (95% CI) unless otherwise indicated. 3. * refers to multiplication.

**Table 4 ijerph-16-00575-t004:** Multivariable ^1^ adjusted analysis of the association between EAFH and MetS ^2^ and its components for each age group.

Indices		Females				Male ORs			
		Never	Sometimes	Often	*p* Trend	Never	Sometimes	Often	*p* Trends
MetS ^2^	Young ^3^	1	0.567(0.364, 0.884) ^4^	0.315(0.143, 0.697)	0.000	1	0.877(0.569, 1.350)	0.802(0.448, 1.436)	0.367
	Middle aged	1	1.060(0.704, 1.596)	1.514(0.766, 2.991)	0.308	1	1.826(1.252, 2.664)	2.725(1.632, 4.550)	0.000
	Elderly	1	1.336 (0.790, 2.260)	1.233(0.384, 3.958)	0.298	1	1.896(1.117, 3.216)	1.561 (0.478, 5.095)	0.031
Component of metabolic syndrome									
High serum TGs	Young	1	0.615(0.423, 0.894)	0.439(0.244, 0.788)	0.000	1	1.183(0.843, 1.660)	1.134(0.739, 1.740)	0.356
	Middle aged	1	1.130(0.789, 1.618)	1.293(0.685, 2.444)	0.320	1	1.414(1.014, 1.972)	1.610(0.993, 2.610)	0.009
	Elderly	1	0.893(0.535, 1.492)	0.991 (0.306, 3.205)	0.737	1	0.958(0.558, 1.645)	0.557(0.163, 1.900)	0.446
Low HDL	Young	1	1.191(0.875, 1.620)	1.301(0.860, 1.967)	0.124	1	1.551(1.031, 2.334)	1.435(0.841, 2.448)	0.036
	Middle aged	1	1.087(0.764, 1.547)	1.045(0.553, 1.975)	0.689	1	1.468(0.958, 2.248)	1.609(0.894, 2.895)	0.027
	Elderly	1	0.753(0.441, 1.287)	0.665 (0.187, 2.372)	0.239	1	0.949(0.476, 1.890)	0.000(0.000, 0.000)	0.174
Abdominal adiposity	Young	1	0.745(0.520, 1.066)	0.569(0.330, 0.981)	0.014	1	0.853(0.573, 1.271)	1.260(0.769, 2.063)	0.711
	Middle aged	1	1.685(1.192, 2.383)	2.216(1.223, 4.016)	0.000	1	1.494(0.990, 2.106)	2.591(1.576, 4.257)	0.000
	Elderly	1	1.755(1.086, 2.837)	1.676(0.538, 5.226)	0.024	1	1.695(0.997, 2.884)	1.644(0.517, 5.234)	0.056
Elevated blood pressure	Young	1	0.386(0.253, 0.590)	0.292(0.145, 0.588)	0.000	1	0.522(0.364, 0.748)	0.718(0.456, 1.129)	0.005
	Middle aged	1	0.825(0.559, 1.218)	0.450(0.193, 1.051)	0.044	1	1.008(0.719, 1.412)	1.771(1.084, 2.894)	0.064
	Elderly	1	2.462(1.505, 4.026)	2.675(0.812, 8.806)	0.000	1	2.045(1.241, 3.370)	4.504(1.416, 14.324)	0.000
Impaired fasting glucose	Young	1	0.534(0.351, 0.812)	0.374(0.190, 0.735)	0.000	1	0.577(0.378, 0.880)	0.396(0.207, 0.760)	0.000
	Middle aged	1	1.114(0.756, 1.643)	1.336(0.674, 2.648)	0.344	1	1.087(0.750, 1.577)	2.257(1.371, 3.715)	0.005
	Elderly	1	1.299(0.780, 2.164)	0.902(0.257, 3.171)	0.524	1	1.380(0.822, 2.317)	1.668(0.600, 4.636)	0.126

Notes: 1. Adjusted for educational level (primary, middle and high), logarithm of income, smoking (yes/no), drinking (currently drinking or not), physical activity (light, moderate, and heavy), localization (urban or rural; north or south), total energy intake, and fat share. 2. Metabolic syndrome was defined as the presence of three or more of the following components: (1) abdominal adiposity (WC ≥102 cm in men and ≥88 cm in women; (2) low serum HDL-cholesterol <40 mg/dL for men and <50 mg/dL for women); (3) high serum triglyceride levels (>150 mg/dL); (4) elevated blood pressure (SBP ≥130mmHg or DBP ≥85 mmHg); and (5) abnormal glucose homeostasis (fasting plasma glucose level ≥100 mg/dL). 3. Young is defined as ≤45 years; Middle aged is defined as >45 years and ≤60 years; Elderly is defined as >60 y. 4. Values are ORs (95% CI), unless otherwise indicated.

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
