# Peer review of "Does Eating-Away-from-Home Increase the Risk of a Metabolic Syndrome Diagnosis?"

_ijerph, 2019, doi:10.3390/ijerph16040575_

Round 1

Reviewer 1 Report

Dear sir, thank you to select me to review manuscript Wang H et al. Does eating-away-from-home increase the risk of getting metabolic syndrome? The authors analyzed 2441 female and 2077 male. Authors concluded, that eating-away-from-home  is associated with higher risk of Metabolic syndrome in older men and it has protective role in Metabolic syndrome in young women.

Author Response

Dear sir, thank you to select me to review manuscript Wang H et al. Does eating-away-from-home increase the risk of getting metabolic syndrome? The authors analyzed 2441 female and 2077 male. Authors concluded, that eating-away-from-home  is associated with higher risk of Metabolic syndrome in older men and it has protective role in Metabolic syndrome in young women. Responses: Thank you very much for your positive evaluation of our paper. We have revised our paper according to the comments proposed by you and other three reviewers. Particularly, we adopted the criteria of central obesity for Asian people and re-conducted all empirical analysis. In addition, we also revised the discussion section and edited the writing. We hope the revised manuscript meet your expectation.

Reviewer 2 Report

Using data from the Chinese Health and Nutrition Survey (2009) among Chinese residents aged more than 18 years old, the authors conducted a study to investigate the association between metabolic syndrome (MetS) and EAFH (eating-away-from-home) at different ages. EAFH increased the risk of MetS for middle aged male (45-60 years), but decreased the risk of MetS for young female (<45 years), and so on. These varied effects of EAFH indicated a careful analysis of the meaning of EAFH is warranted, such as who is likely to eat out at different age-sex groups. For working class, maybe those in high SES and certain lifestyle tends to eat out. For elderly, maybe those without caregiver tend to eat out. Authors did not proceed with an in-depth analysis, but made a bold conclusion that heterogeneous target strategies should be considered for preventing MetS in different subpopulation. The reviewer feel that this conclusion is not helpful for advancing the public health knowledge. My suggestion to the authors is to make further analysis and understand the meaning of EAFH in different age-sex groups and make more constructive conclusions.

There are some other issues which need further modifications.

1.     The authors used the criteria of central obesity with more than 102 cm for men and more than 88 cm for women. This criteria may not fit for the Chinese population. The authors should consider the cutoff values of waist circumference with 2125px for man and 75.0 cm for female in Chinese in this study.

2.     The territory of CHINA is vast and the dietary culture may be different among areas. Has the association between EAFH and metabolic syndrome varied by geographical regions? The authors may provide the association stratified by urban and rural and by north and south.

Author Response

Using data from the Chinese Health and Nutrition Survey (2009) among Chinese residents aged more than 18 years old, the authors conducted a study to investigate the association between metabolic syndrome (MetS) and EAFH (eating-away-from-home) at different ages. EAFH increased the risk of MetS for middle aged male (45-60 years), but decreased the risk of MetS for young female (<45 years), and so on. These varied effects of EAFH indicated a careful analysis of the meaning of EAFH is warranted, such as who is likely to eat out at different age-sex groups. For working class, maybe those in high SES and certain lifestyle tends to eat out. For elderly, maybe those without caregiver tend to eat out. Authors did not proceed with an in-depth analysis, but made a bold conclusion that heterogeneous target strategies should be considered for preventing MetS in different subpopulation. The reviewer feel that this conclusion is not helpful for advancing the public health knowledge. My suggestion to the authors is to make further analysis and understand the meaning of EAFH in different age-sex groups and make more constructive conclusions.

Responses: Thank you very much for your suggestion. To provide an in-depth analysis of the mechanism behind the association between MetS and EAFH, we further sum up the frequency of eating breakfast, lunch, dinner and snack away from home for each age-sex groups. Results are presented in Table S3 in the supplementary file. We find that middle aged male had significant higher frequency of EAFH compared with middle aged female. That could be explained by the fact that Chinese men engaged in more social activities which usually take place in restaurants, and they tend to drink a lot (see Table 1) and eat a lot during these meals (Tian et al., 2016). Regarding female, as young females, are usually more self-aware and more subject to peer pressure and preferences for thinness (Min et al., 2018); thus they are less likely to be affected by EAFH. More importantly, young females who have higher frequency of EAFH are usually those who have higher social-economic-status, and they are well-educated (see Table 1), have decent jobs (which can be reflected by the lower physical activity level shown in Table 1), and have a more healthy lifestyle (less likely to smoke and drink as shown in Table), all of which can contribute to a lower risk of MetS. In addition, previous study also found that EAFH contributed to a higher BMI for Chinese men, while it is not associated with BMI increase for women, and they even find some evidence that women who eat lunch away from home more often have lower BMI (Tian et al., 2016).

 There are some other issues which need further modifications.

 1.     The authors used the criteria of central obesity with more than 102 cm for men and more than 88 cm for women. This criteria may not fit for the Chinese population. The authors should consider the cutoff values of waist circumference with 2125px for man and 75.0 cm for female in Chinese in this study.

Responses: Thank you very much for figuring out this issue. We agree that the international standard might not fit for Chinese people. We have adopted the standard for Asian people in the revised manuscript. The cutoff values of waist circumference are 2250px for man and 85 for female respectively. We conducted all empirical analysis again, and found that the results were consistent with the old one.

 2.     The territory of CHINA is vast and the dietary culture may be different among areas. Has the association between EAFH and metabolic syndrome varied by geographical regions? The authors may provide the association stratified by urban and rural and by north and south.

Responses: We agree that the dietary culture varies across different regions in China, which may affect the association between EAFH and MetS. We thus follow your suggestion and investigate the association separately for urban and rural and north and south. Results are presented in Table 2 in supplementary file.

Reviewer 3 Report

-Why not include the supplementary file (Table 1 and figure1) in the texte at the appropriate site ?

-In the abstract you mentioned "a significant protective role in MetS for young females", however in the four important limitations (and you cite it well) there is no causal relationship.

-also line 307 is wrong formulated since in the young generation females will not go up in the rising prevalence of MetS.

Author Response

-Why not include the supplementary file (Table 1 and figure1) in the texte at the appropriate site ?

Responses: Thank you for your suggestion. The reason for not including the supplementary file in the text is due to the limitation of number of table and figure in the main text. We thus only select three tables and two figures which we believe is most important in the main text, and present other results as supporting evidence in the supplementary file.

-In the abstract you mentioned "a significant protective role in MetS for young females", however in the four important limitations (and you cite it well) there is no causal relationship.

Responses: Thank for your careful reading. We agree that this study only detects the association between EAFH and MetS. Causality investigation needs more complicated empirical design and data with better quality such as longitudinal data. We have carefully checked the whole paper and revised all such expression to avoid misunderstanding.

-also line 307 is wrong formulated since in the young generation females will not go up in the rising prevalence of MetS.

Responses: Thank for very much for your comment. We have deleted “particularly for young generation” from the sentence to make the statement consistent with our results.

Reviewer 4 Report

Materials and Methods:
Despite of some reasons, the data used in the study was too old. In my opinion, there had been various changes in demographic, health-related characteristics in Chinese population for almost 10 years (2009-2018). This limitation must be discussed more in-detail. 

Not only personal health history, but also family history is one of the critical factors can be affect one's health status(having chronic disease). 

The flow chart is better to demonstrate the study population.

Methods: This cross-sectional study cannot be guarantee causality. 

Discussion:
The result of this study is quite interesting(gender and age difference by EAFH), but the discussion part is too shallow.  More discussion about the results and more references will be needed.  

Author Response

Materials and Methods:
Despite of some reasons, the data used in the study was too old. In my opinion, there had been various changes in demographic, health-related characteristics in Chinese population for almost 10 years (2009-2018). This limitation must be discussed more in-detail. 

Responses: Thank you for your comment. We agree that in a rapid growing country like China, there may have various changes in demographic and health-related characteristics, which may lead to a different association between EAFH and MetS. We have briefly discussed the limitation of the data in the discussion section, and call for new dataset to evaluate our finding.

Not only personal health history, but also family history is one of the critical factors can be affect one's health status(having chronic disease). 

Responses: Thank you for your suggestion. We generate a new variable “disease” to identify whether any member in the family have been diagnosed as hypertension, diabetes, and stroke, and take it as a proxy variable to capture the family history of chronic disease. We add it in the model and re-conduct all empirical analysis. New results are presented in the revised manuscript. In general, we do not find so much difference compared with previous results.

The flow chart is better to demonstrate the study population.

Responses: Thank you for your suggestion. We have added a flow chart in the supplementary file to demonstrate the sample selection procedure.

Methods: This cross-sectional study cannot be guarantee causality. 

Responses: We agree that this study only detects the association between EAFH and MetS. Causality investigation needs more complicated empirical design and data with better quality such as longitudinal (panel) data. We have carefully checked the whole paper and revised all expression of causality to avoid misunderstanding.

Discussion:

The result of this study is quite interesting(gender and age difference by EAFH), but the discussion part is too shallow.  More discussion about the results and more references will be needed.  

Responses: Thank you for your suggestion. We have read more literature on relevant topics and substantially revised the discussion section. In particular, we added more discussion on the different results found between male and female. We hope the new manuscript meet your expectation.

Round 2

Reviewer 2 Report

Since the manuscript is generated from nutrition survey data, it is easy to check the quality of foods selected by females and by males. If females tend to choose nutrient-dense foods, but males do not; one can make a meaningful conclusion. Currently, There is no check on the thoroughness of adjustment for confounders, it is uncertain whether the differential findings for men and for women is real.

I do not think a manuscript providing this kind of conclusion is productive and positive to health promotion and disease prevention.

Reviewer 4 Report

Despite of all the merits of this study, still there are some issues in the study. The major problem is EAFH(frequency) in not only the affecting factor with MetS prevalence. I think that more detail analysis is need to investigate(e.g., menu, nutrition fact etc.).